# X-rays-Induced Bystander Effect Consists in the Formation of DNA Breaks in a Calcium-Dependent Manner: Influence of the Experimental Procedure and the Individual Factor

**DOI:** 10.3390/biom13030542

**Published:** 2023-03-16

**Authors:** Juliette Restier-Verlet, Aurélie Joubert, Mélanie L. Ferlazzo, Adeline Granzotto, Laurène Sonzogni, Joëlle Al-Choboq, Laura El Nachef, Eymeric Le Reun, Michel Bourguignon, Nicolas Foray

**Affiliations:** 1INSERM U1296 unit “Radiation: Defense/Health/Environment” Centre Léon-Bérard, 69008 Lyon, France; 2Department of Biophysics and Nuclear Medicine, Université Paris Saclay Versailles St Quentin en Yvelines, 78035 Versailles, France

**Keywords:** bystander effect, DNA double-strand breaks, calcium, radiosensitivity

## Abstract

Radiation-induced bystander effects (RIBE) describe the biological events occurring in non-targeted cells in the vicinity of irradiated ones. Various experimental procedures have been used to investigate RIBE. Interestingly, most micro-irradiation experiments have been performed with alpha particles, whereas most medium transfers have been done with X-rays. With their high fluence, synchrotron X-rays represent a real opportunity to study RIBE by applying these two approaches with the same radiation type. The RIBE induced in human fibroblasts by the medium transfer approach resulted in a generation of DNA double-strand breaks (DSB) occurring from 10 min to 4 h post-irradiation. Such RIBE was found to be dependent on dose and on the number of donor cells. The RIBE induced with the micro-irradiation approach produced DSB with the same temporal occurrence. Culture media containing high concentrations of phosphates were found to inhibit RIBE, while media rich in calcium increased it. The contribution of the RIBE to the biological dose was evaluated after synchrotron X-rays, media transfer, micro-irradiation, and 6 MeV photon irradiation mimicking a standard radiotherapy session: the RIBE may represent less than 1%, about 5%, and about 20% of the initial dose, respectively. However, RIBE may result in beneficial or otherwise deleterious effects in surrounding tissues according to their radiosensitivity status and their capacity to release Ca^2+^ ions in response to radiation.

## 1. Introduction

Since more than 100 years, a considerable body of data suggests the existence of biological effects in non-targeted cells situated in close proximity to irradiated ones [1,2]. Even if these effects are not necessarily generated by a single cause, and despite the fact that their molecular bases are not fully identified yet, radiobiologists have currently described them under the single term of radiation-induced bystander effect (RIBE) [3,4,5,6]. RIBE has been documented with various endpoints like gene mutation induction, DNA damage, apoptosis, and/or malignant transformation [7,8,9,10,11,12,13,14] and by applying various experimental procedures like cytoplasmic irradiations [9,15], exposures to alpha-particles [6,16,17], charged particle microbeams [9,18] and irradiated culture medium transfers [19,20]. To date, it has been difficult to develop a mechanistic and unified model of RIBE without identifying the molecular and cellular features that are common to all these experimental procedures. Medium transfer and micro-irradiation approaches are the most current methodologies to investigate RIBE. However, most micro-beam experiments have involved heavy ions or alpha particles, whereas most medium transfer experiments have been performed with X-rays [9,15]. Hence, in order to avoid any methodological bias and technical artifact, a major step to elucidate the RIBE mechanisms would be to apply both medium transfer and micro-irradiation by using the same type of radiation. Synchrotrons are accelerators of electrons that, after being decelerated, produce high-flue nced X-rays. Such high-fluence synchrotron X-rays permit to irradiate cells by micro-tracks at a non-negligible dose rate and by covering a sufficiently homogenous irradiation. Synchrotron X-rays represent therefore a real opportunity to investigate RIBE by applying medium transfer and micro-irradiation approaches using the same type of radiation [21,22,23,24,25].

In parallel to this technical opportunity, it has been demonstrated that radiation-induced DNA double-strand breaks (DSB), one of the major DNA damages induced by ionizing radiation, can be determined by the phosphorylation of H2AX histone (γH2AX) which forms nuclear foci at the DSB sites. These γH2AX foci are easily quantifiable by immunofluorescence [26]. The presence of γH2AX foci in RIBE cells has already been reported [20,27]. In literature, the formation of late DSB appears to be a major feature of RIBE. However, it must be stressed that nuclear γH2AX foci reflect a subset of radiation-induced DSB, those that are recognized by the non-homologous end-joining (NHEJ) pathway, predominant in mammalian quiescent cells. Hence, some other DSB can be produced by RIBE but may not be revealed by γH2AX foci; some other biomarkers are therefore needed to complete the molecular characterization of RIBE to determine whether other DNA repair pathways than NHEJ are involved in RIBE. The nuclease MRE11 forms nuclear foci after irradiation and produces a subset of DSB that are not managed by NHEJ [28]. Since the γH2AX foci/MRE11 foci ratio may condition the individual response to radiation [29], the nature of the DSB involved in RIBE raises also the question of the influence of the individual factor in this radiobiological effect.

Lastly, in parallel to the generation of some late DSB and the potential involvement of the MRE11 nuclease, another current observation described in the literature about RIBE is the radiation-induced extracellular flux of Ca^2+^ ions that may propagate among the RIBE cells [30]. Besides, lindane, an inhibitor of gap-junctions, has been found to inhibit RIBE, which highlighted the role of intercellular connections in RIBE. However, it is noteworthy that lindane also disturbs calcium homeostasis [11,31]. Furthermore, the formation of radiation-induced DSB and the activation of MRE11 nucleases have also been shown to depend on Ca^2+^ ions [32,33]. Hence, the amount and the availability of calcium before irradiation appear to be two key elements for the occurrence and extent of RIBE.

In this study, we attempted to provide a unified mechanistic model of RIBE that would be independent of the experimental protocol. To achieve this aim, as a first step, both medium transfer and micro-irradiation approaches were applied with synchrotron 50 keV X-rays to a human radioresistant fibroblast cell line by using γH2AX foci as molecular endpoints. As a second step, the role of the MRE11 nuclease in RIBE and the influence of calcium and its contribution to the total dose were evaluated.

## 2. Materials and Methods

### 2.1. Cell Lines

Four human untransformed cutaneous fibroblasts cell lines were chosen to avoid any bias linked to genomic stability. Fibroblasts were routinely cultured at 37 °C in 5% CO_2_ humid conditions as monolayers with Dulbecco’s modified Eagle’s minimum medium (DMEM) (Gibco-Invitrogen-France, Cergy-Pontoise, France), supplemented with 20% fetal calf serum, penicillin, and streptomycin. In all the experiments, fibroblasts were maintained in the plateau phase of growth (95–99% in G0/G1) to overcome any cell cycle effect. The radioresistant 1BR3 cell line was derived from an apparently healthy donor, and its radiobiological features were published elsewhere [34,35,36]. The 14CLB and 200CLB cell lines were originated from a donor who showed an adverse tissue reaction after anti-cancer radiotherapy and from a donor who showed spontaneous breast cancer treated by surgery, respectively. Finally, the GM01142 cell line was derived from a donor suffering from retinoblastoma syndrome and purchased from Coriell Cell Repositories (Camden, NJ, USA) [37]. These cell lines belong to the “COPERNIC” collection, managed by our lab and approved by the regional Ethical Committee. The COPERNIC cell lines were declared under the numbers DC2008-585, DC2011-1437 and DC2021-3957 to the Ministry of Research. Our radiobiological database was protected under the reference number IDDN.FR.001.510017.000.D.P.2014.000.10300 [36].

### 2.2. Anti-Oxydative and Calcium-Dependent and Independent Drugs

*N*-acetyl-l-cysteine (NAC) (#A7250), NaCl, 0.9% (physiological serum, #S9888), Na_2_HPO_4_ (#342483), and CaCl_2_ (#C1016) were purchased from Sigma-Aldrich (Saint-Quentin-Fallavier, France). Calcium and magnesium free-phosphate-buffered-saline solution (PBS or PBBwCa; #14040091) and calcium-free PBS (PBSw/oCa; #14190144) were purchased from Thermo Fisher Scientific (Waltham, MA, USA).

### 2.3. High-Energy Photons Irradiations

High-energy photon irradiations were performed with a 6 MeV photon medical irradiator (SL 15 Philips) (dose-rate: 6 Gy.min^−1^) at the anti-cancer Centre Léon-Bérard (CLB) (Lyon, France). Dosimetry was certified by the radiophysicists of the CLB [35,36].

### 2.4. Synchrotron Low-Energy X-rays Irradiation

Experiments involving synchrotron X-rays were performed at 50 keV at the biomedical ID17 beamline of the European Synchrotron Radiation Facility (Grenoble, France). Irradiation features were detailed elsewhere [38,39,40]. Micro-irradiations were performed with a set-up made of two 100-µm-wide parallel micro-tracks separated by 300 µm. Dosimetry was routinely checked by radiochromic films [40].

### 2.5. Medium Transfer Approach

If another number of cells is not indicated, 10^6^ cells were seeded in 25 cm^2^ flasks to reach confluence (Appendix A). After 3 days of incubation, the confluent donor cells were irradiated in the indicated medium and maintained for an incubation period of t1 (10 min–4 h) at 37 °C. The medium was then removed and centrifuged for 2 min (1500 rpm, 200× *g*, 37 °C) and the surpernatant was then applied to unirradiated confluent autologous receptor cells and maintained for an incubation period of t2 (1–24 h) at 37 °C. The immunofluorescence was then processed.

### 2.6. Free Calcium Quantification

The content of free calcium was determined by atomic absorption spectrometry using an air-acetylene flame. Confluent cells were scraped, centrifuged for 2 min (1500 rpm, 200× *g*, 4 °C) and the surpernatant was removed. Cells were then washed, and two-fold diluted in a solution containing 2.5% HCl, 0.65% lanthane oxide, and 0.005% CsCl. Calcium concentration was obtained after calibration with Ca standards [41].

### 2.7. Immunofluorescence

The immunofluorescence protocol was described elsewhere [42,43]. Briefly, cells fixed in paraformaldehyde for 10 min at room temperature and were permeabilized in 0.5% Triton X-100 solution for 5 min at 4 °C. Primary and secondary antibody incubations were performed for 40 and 20 min at 37 °C, respectively. The anti-*γH2AX^ser139^* (#05636; Upstate Biotechnology-Euromedex, Mundolsheim, France) and the anti-*MRE11* (#56211; QED Bioscience, San Diego, CA, USA) antibodies were used at 1:800 and 1:100, respectively. Incubations with anti-mouse fluorescein (FITC) and rhodamine (TRITC) secondary antibodies were performed at a ratio of 1:100 at 37 °C for 20 min. Slides were mounted in 4′,6′ Diamidino-2-Phenyl-indole (DAPI)-stained Vectashield (Abcys, Paris, France). Nuclear foci were examined with Olympus fluorescence microscope (Olympus France S.A.S, Rungis, France). Foci scoring procedure applied here has received the certification agreement of CE mark and ISO-13485 quality management system norms. Our foci scoring procedure also developed some features that are protected in the frame of the Soleau Envelop and patents (FR3017625 A1, FR3045071 A1, EP3108252 A1) [29,36,43,44]. One hundred nuclei were examined per condition and per experiment to score the standard nuclear γH2AX and MRE11 foci. The size of *γH2AX* foci was controlled by the Cells software program linked to the Olympus microscope. All the foci whose surface was lower than 1 µm^2^ was considered as “tiny” foci. The distance from the micro-tracks was determined by moving the micrometric wheel of the microscope and by verifying the position based on the average size of cell nuclei.

### 2.8. Statistical Analysis

Statistical significance between data points was verified with the one-way ANOVA test, and the statistical significance between two curves was verified with the non-parametric Friedman test. Statistical analysis was performed by using Kaleidagraph v4 (Synergy Software, Reading, PA, USA). Data fitting was performed with the indicated mathematical formulas.

## 3. Results

### 3.1. RIBE Induced by a Medium Transfer Approach

The radioresistant 1BR3 fibroblasts were irradiated in the plateau phase of growth with monochromatic 50 keV synchrotron X-rays in DMEM medium. The DMEM medium was maintained for t1 post-irradiation. The DMEM medium was thereafter applied to unirradiated autologous receptor cells for t2. The RIBE DSB revealed by nuclear γH2AX foci was examined in receptor cells with t1 ranging from 10 min to 4 h and t2 ranging for 1 to 24 h. An incubation of donor cells for ten min (t1) after irradiation followed by an incubation of receptor cells for 4 h (t2) provided the largest number of γH2AX foci per cell. This conclusion was reached regardless the initial dose (Figure 1A).

We examined thereafter whether the number of irradiated donor cells strongly affects the γH2AX data in the optimal medium transfer conditions defined above, namely t1 = 10 min and t2 = 4 h. A large range of irradiated donor cells (from 10^2^ to 10^7^ cells) was tested. No significant RIBE was observed with 10^2^ to 10^4^ irradiated cells. The number of γH2AX foci per cell obeyed a sigmoidal function of the number of irradiated cells with a threshold found at 10^4^ irradiated cells (Figure 1B). The shape of the sigmoidal curve changed with the incident dose: the higher the dose, the higher the RIBE extent. In our conditions, a maximal RIBE was observed at doses higher than 2 Gy and with more than 10^6^ irradiated cells (Figure 1B). Finally, in all these experiments, cells with γH2AX-positive signals were dispatched heterogeneously among the receptor cells, but the maximal number of γH2AX foci per cell never exceeded 5 and generally averaged between 1 and 2 γH2AX foci per cell, which would correspond to a standard exposure ranging between 25 and 50 mGy i.e., less than 1% initial dose [45].

Altogether, our findings suggested that the medium transfer approach with monochromatic 50 keV synchrotron X-rays induces RIBE that results in DSB recognized by NHEJ (i.e., γH2AX positive) occurring from 10 min post-irradiation and remaining persistent up to 4 h post-irradiation. Such RIBE was found to be dependent on the dose and on the number of irradiated cells and may be equivalent to a dose of some tens of mGy. It is noteworthy that no γH2AX foci was observed when transfer medium was irradiated in the absence of cells, supporting that the occurrence of RIBE requires the presence of donor cells during irradiation (data not shown). Lastly, it is noteworthy that no apoptotic body was observed in RIBE fibroblasts.

### 3.2. RIBE Induced by a Micro-Irradiation Approach

The formation of RIBE DSB was examined in the vicinity of two 100-µm-wide parallel 50 keV synchrotron X-ray micro-tracks separated by 300 µm. Four initial doses were applied to the radioresistant 1BR3 fibroblasts: 1, 2, 5 and 10 Gy. Ten minutes after exposure to 10 Gy X-rays, irradiated cells in both micro-tracks were easily identifiable by an intense broad γH2AX staining in which the γH2AX were not distinguishable. Indeed, by considering that a dose of 1 Gy X-rays induces about 40 DSB per human diploid non-transformed fibroblast, 10 × 40 DSB/Gy = 400 DSB were expected in each cell [46]. In our conditions, each γH2AX foci can be distinguishable in a fibroblast nucleus when their number does not exceed 100 [46] (Figure 2). In the cells situated at the outer edges of the micro-tracks, some γH2AX foci may be identifiable (Figure 2). I n the cells situated at more than 10 μm from the micro-tracks, some γH2AX foci were randomly dispatched in the nuclei, irrespectively of the orientation of the synchrotron X-rays micro-tracks. Considering the mean-free path of 50 keV X-rays, the DSB revealed by the γH2AX foci in cells situated at more than 10 μm from the micro-tracks cannot be physically induced by synchrotron X-rays (Figure 2). Furthermore, since more than 95% cells are in G0/G1 phase (Appendix A), and since irradiated cells into the micro-tracks represented about 1% of the whole cell population seeded in the microscope slides, the DSB observed outside from the micro-tracks cannot be due to cell migration. Hence, our observations supported that the synchrotron X-rays micro-irradiation set-up applied here can produce a significant RIBE.

The number of γH2AX foci per cell decreased as a function of the distance from the micro-tracks. The mathematical function that links the number of γH2AX foci and the distance from the micro-tracks was also found dependent on the post-irradiation time (Figure 3). Particularly, the γH2AX foci observed in RIBE cells appeared early (10 min) after irradiation. In our conditions, the maximal number of γH2AX foci was reached at 1 h post-irradiation. The larger distance from the micro-tracks at which some γH2AX foci were still observed was reached at 4 h post-irradiation (Figure 3A,B). Similar observations were made with the other doses tested (data not shown for 1 and 5 Gy).

By considering that a dose of 1 Gy X-rays induces about 40 DSB per human diploid non-transformed fibroblast [46], the numbers of γH2AX foci can be expressed in Gy-equivalent by dividing them by 40: considering the slope of the curve shown in Figure 3C,D, the RIBE appeared to be equivalent to about 5.6% of the incident dose (Figure 4A).

The distance from the micro-track at which the number of γH2AX foci was not nil whatever the post-irradiation time tested was 10 μm (Figure 3): the repair kinetics of DSB assessed at such a distance from the micro-track reflects the repair rate of the RIBE DSB induced at 10 µm from the micro-track. Such repair kinetics were obtained by plotting the number of γH2AX foci against the corresponding post-irradiation time (Figure 4B). Interestingly, the resulting DSB repair kinetics appeared significantly different from the DSB repair kinetics found in the same radioresistant control fibroblasts but after a standard (direct) irradiation [29] (*p* < 0.01) (Figure 4B). These findings suggest that the RIBE DSB are more slowly repairable than those induced by a single irradiation performed in standard conditions (Figure 4B). Such a statement remains true for 1, 5, and 10 Gy (data not shown).

Lastly, in order to better document the speed of the RIBE after a 50 keV synchrotron X-rays micro-irradiation, the maximal distance reached by γH2AX foci was plotted against the corresponding post-irradiation time (Figure 4C). The resulting curve elicited two phases of appearance/disappearance of the RIBE DSB: an increasing pseudo-linear slope reflecting the propagation of DSB in the RIBE cells and a decreasing pseudo-exponential phase reflecting the repair of DSB. In the first phase, the RIBE reached 300 µm from the micro-tracks in the first 4 h post-irradiation (i.e., with average speed of 75 µm/h). In the second phase, the RIBE described a decrease in the number of DSB and an inverse progression of 200 µm from 4 to 24 h post-irradiation (i.e., with an average speed of 10 µm/h) (Figure 4C).

### 3.3. The “Tiny” γH2AX Foci: A Common Feature of RIBE?

Some γH2AX foci observed in the RIBE cells, whether after a medium transfer or a micro-irradiation showed a specific pattern: among the “standard” γH2AX foci, some foci were found to be “tiny” (less than 1 µm^2^) i.e., significantly smaller than those observed routinely in non-RIBE conditions (Figure 5A). It must be stressed that these tiny foci were not included in the foci scoring during the experiments described above (see Materials and Methods). In our conditions, such a tiny γH2AX foci pattern has also been encountered when DSB and DNA single-strand breaks (SSB) are present concomitantly, i.e., when chromatin is relaxed in the presence of DSB (Figure 5A). This is notably the case after high doses of UV through the base damage conversion into SSB and after a H_2_O_2_ treatment that produces SSB and DSB concomitantly (Figure 5A). Some tiny γH2AX foci may be also observed 10 min after 2 Gy in human fibroblasts derived from radiosensitive donors, likely due to the presence of both SSB and DSB (Figure 5A). However, in this case, the tiny γH2AX foci are generally less persistent than the standard ones and disappear about 2 h after 2 Gy X-rays, reflecting the short repair half-time of SSB (about 20 min in human fibroblasts) and the longer repair half-time of DSB (about 1 h in human fibroblasts) [46]. SSB are not detected directly via the γH2AX signals, but they trigger the chromatin decondensation that disperses several of the H2AX molecules concentrated around one DSB site: a “standard” γH2AX focus revealing one DSB becomes a cloud of “tiny” foci revealing some SSB around (Figure 5A).

Whether with the medium transfer or the micro-irradiation approach, the tiny γH2AX foci were observed together with standard ones, suggesting a continuous production of SSB with the presence of DSB during RIBE. Oxidative stress generally activates TFHII components and nucleases like MRE11 [48]. Activated MRE11 nuclease has been shown to be responsible for the formation of SSB, whether after X-rays or UV [49]. The ATM kinase has been shown to phosphorylate MRE11 foci, which triggers the formation of nuclear MRE11 foci. The MRE11 foci were also found early after exposure to UV [50]. Whether with the medium transfer or the micro-irradiation approach, MRE11 foci appeared early after irradiation (10 min) in RIBE cells, and their numbers increased during the first hour post-irradiation. Thereafter, the number of MRE11 foci in the RIBE cells decreased to zero at 24 h post-irradiation. Interestingly, the kinetics of MRE11 foci appeared similar to that of γH2AX foci, with two phases composed of a rapid appearance rate and a slow disappearance rate (Figure 4 and Figure 5B).

Altogether, our findings support strong analogies between the RIBE generated by the medium transfer and the micro-irradiation approach. The RIBE signals can be persistent for 4 h post-irradiation and observable up to 300 µm from the radiation impact.

### 3.4. Culture Medium Composition and RIBE

All the above experiments were performed in DMEM medium supplemented with bovine serum. Since RIBE has been found to be strongly dependent on specific chemical species [51], we examined whether the chemical composition of the DMEM medium impacts RIBE. As a first step, medium transfer was performed with physiological serum (NaCl, 0.9%) and calcium and magnesium free-phosphate-buffered saline solution (PBSw/oCa), without bovine serum or antibiotics. By applying 1, 2, 5 and 10 Gy synchrotron X-rays with t1 = 10 min and t2 = 4 h to one million 1BR3 fibroblasts in physiological serum, the number of γH2AX foci appeared to be similar to that observed with DMEM (Figure 6A,B). Conversely, no significant RIBE was observed with PBSw/oCa and with physiological serum combined with phosphates (Na_2_HPO_4_), while PBS with calcium allowed the occurrence of a significant RIBE (Figure 6A,B). These findings suggested that: (1) physiological serum (NaCl, 0.9%) is the minimal medium allowing RIBE. Neither bovine serum nor antibiotics significantly impact upon the RIBE occurrence (data not shown). (2) PBSw/oCa is the minimal medium preventing RIBE. PBSw/oCa contains more than 1 mM KH_2_HPO_4_ and 8 mM Na_2_HPO_4_, suggesting that phosphates are more abundant in PBSw/oCa than in any other medium tested (Table 1). Phosphates are known for their affinity with calcium. By using PBS with calcium (PBSwCa; 0.91 mM CaCl_2_), the number of γH2AX foci per cell was found to be larger than with PBSw/oCa (*p* < 0.01) but remained statistically lower than with DMEM that contains higher calcium concentrations (1.8 mM CaCl_2_) (*p* < 0.001) (Table 1) (Figure 6A).

Interestingly, when a powerful anti-oxidative drug like N-acetylcysteine (NAC) was applied to receptor cells, less γH2AX foci appeared. Particularly, 50 mM NAC resulted in 5-fold decreased yield of γH2AX foci, suggesting that NAC influences the production of late DSB by preventing the formation of RIBE-induced reactive oxygen species (ROS) and DNA damage (Figure 6C). Similarly, when the micro-irradiation approach was applied in PBSw/oCa, the RIBE did not occur (data not shown), supporting again the dependence of RIBE vis-à-vis ROS, phosphates and availability of calcium.

### 3.5. Calcium and RIBE

A number of authors have reported the existence of a radiation-induced release of intracellular Ca^2+^ ions between 10 min and 1 h post-irradiation [30,52]. Literature and our findings prompted us to verify whether RIBE is dependent upon the availability of free Ca^2+^ ions. One million 1BR3 fibroblasts were irradiated (10 Gy) in either PBSw/oCa or physiological serum and the free calcium concentration was determined in irradiated media 10, 30, 60 and 240 min post-irradiation by using flame atomic absorption spectrometry [41] (Figure 7). With the physiological serum, the free Ca^2+^ ions concentration increased 10 min post-irradiation and remained higher than controls up to 60 min post-irradiation. Conversely, no significant increase was found when cells were irradiated in PBSw/oCa (Figure 7). Our findings suggest that the intracellular Ca^2+^ ions release may diffuse from irradiated cells into extracellular medium. The presence of phosphates like those containing in PBSw/oCa may contribute to prevent the radiation induced Ca^2+^ ions release (Figure 7A,B).

### 3.6. Contribution of RIBE to the Radiation Dose-Effect

Our findings suggest that RIBE represents a contribution to the biological dose that depends on the irradiation protocol: with monochromatic 50 keV synchrotron X-rays, the RIBE induced by medium transfer and micro-irradiation represents less than 1% and about 5% of the initial dose, respectively. In order to compare these findings with a more current irradiation scenario, we evaluated the relative contribution to the dose of the Ca-dependent events during conditions of irradiation mimicking a standard anti-cancer radiotherapy session with high-energy photons. To this aim, cells were irradiated with a 6 MeV photons medical irradiator by applying different drugs pre-treatments promoting or inhibiting the radiation induced Ca^2+^ ions release (Figure 8 and Figure 9).

A dose of 2 Gy high-energy photons delivered to the radioresistant 1BR3 cells in DMEM produced about 74 ± 6 γH2AX foci per cell at 10 min post-irradiation. In these conditions, all the γH2AX foci disappeared at 24 h post-irradiation. While the irradiation in physiological serum did not change these values significantly (data not shown), an irradiation performed in PBS produced 60 ± 4 γH2AX foci per cell at 10 min post-irradiation i.e., about 20 γH2AX foci less than with DMEM (*p* < 0.05) (Figure 8A). By expressing γH2AX foci data as percentage of γH2AX foci remaining, such value corresponds to about 75–80% γH2AX foci assessed at 10 min with PBS. In other terms, calcium-dependent events contribute to about 20–25% of the initial dose. Interestingly, by adding CaCl_2_ to PBS, the number of γH2AX foci per cell reached that assessed with DMEM (*p* > 0.2) (Figure 8B). It is noteworthy that, in all these experiments, the tiny γH2AX foci appeared 10 min post-irradiation but disappeared thereafter (data not shown).

### 3.7. Individual Radiosensitivity and RIBE

In the previous experiments, we examined the occurrence and the intensity of RIBE in the untransformed radioresistant 1BR3 fibroblasts that show a survival fraction at 2 Gy (SF2) higher than 60% [28]. In these radioresistant cells, the ATM kinase diffuses rapidly in the nucleus in response to radiation [53]. Once in the nucleus, the ATM kinase can phosphorylate the H2AX variant histone to form the γH2AX foci at the sites of DSB that are recognized by NHEJ [53]. In parallel, the ATM kinase phosphorylates the MRE11 protein, which triggers the formation of MRE11 foci, reflecting the inhibition of the nuclease activity of the MRE11 protein [28]. In radiosensitive cells showing a delay in the radiation-induced ATM nucleo-shuttling (RIANS), some cytoplasmic and overexpressed proteins prevent the RIANS and form multiprotein complexes with ATM in the cytoplasm. As a result, the delay in RIANS may facilitate continuous activation of the MRE11 nuclease [28]. In order to examine the influence of individual radiosensitivity on RIBE, four fibroblast cell lines showing different radiosensitivity and RIANS status were irradiated (2 Gy high-energy photons) in DMEM or PBS and submitted to anti-*γH2AX* and MRE11 immunofluorescence. Foci data were plotted against the post-irradiation time (Figure 9).

With regard to the γH2AX data, the number of foci decreased faster when irradiation was made in PBS than in DMEM (Figure 9A), whatever the radiosensitivity. Furthermore, the γH2AX data assessed 10 min post-irradiation, reflecting the DSB subset recognized by NHEJ were systematically lower in PBS than in DMEM (*p* < 0.01) (Figure 9A).

Interestingly, when the corresponding DMEM and PBS data were plotted together, it appeared that PBS pre-treatment resulted in the disappearance of about 25% γH2AX foci, whatever the radiosensitivity and the post-irradiation time (Figure 9B). These findings support that the inhibition of the radiation induced Ca^2+^ ions release by a culture medium rich in phosphates results in a significant (25%) decrease of the formation of DSB or else of the recognition of DSB by NHEJ (Figure 9B). The extent of such a process occurs independently of the radiosensitivity, likely because it involves chemical reactions common to all the cells. However, the biological consequences of such RIBE are likely to be dependent on radiosensitivity.

With regard to the MRE11 data, all the foci kinetics showed their maximum early after irradiation (10 min or 1 h) when irradiation was performed in DMEM. By contrast, when irradiation was performed in PBS, the shapes of the MRE11 foci kinetics changed drastically and became curvilinear, with a maximal value of MRE11 foci reached later after irradiation (4 or 24 h post-irradiation) (Figure 9C). Interestingly, irradiation of cells performed in PBS also resulted in about twice fewer foci than those observed with DMEM, independently of the radiosensitivity. Since MRE11 foci reflect the inactivation of the nuclease activity of MRE11 by ATM, our findings suggest that the inhibition of the radiation induced Ca^2+^ ion release by the PBS results in limiting the generation of DNA breaks by MRE11 nuclease but delaying its total inactivation. Conversely, the promotion of RIBE amplifies the MRE11 nuclease activity by generating more DNA breaks and earlier.

## 4. Discussion

### 4.1. A Unified Model for RIBE?

While RIBE has been abundantly documented in literature, its molecular bases remain unclear [3,4,5,6,20,54]. Historically, in vitro RIBE was initially observed with alpha-particles, which generated some doubts with regard the biological nature of the RIBE since physical scattering encountered with high LET-particles can induce energy micro-depositions far from their initial impact [3,4,5,6]. Injuries at gap junctions have been also proposed as a common cause of RIBE. However, gap junctions inhibitors were shown to reduce the RIBE generated by alpha particles while they were found inefficient in medium transfer experiments involving X-rays [11,20,55]. Despite such controversies, a temporal analogy in the formation of RIBE DSB was suggested with both approaches, namely micro-irradiations with alpha-particles and medium transfer with X-rays [3,10,55,56]. However, to be relevant, such an analogy should be observed with the same type of radiation As evoked in the Introduction, thanks to their high fluence, synchrotron X-rays represented a real opportunity to perform both medium transfer and micro-irradiations approaches with the same radiation type [21,22,23,24,25].

### 4.2. Chemically Induced DNA Breaks as Major Feature of RIBE?

RIBE DSB differ from DSB induced by X-rays by their late occurrence. With both medium transfer and micro-irradiation approaches RIBE DSB were observed from 10 min to 4 h post-irradiation while the “physically” induced DSB observed after a “direct” irradiation are generated some microseconds after irradiation [57]. High doses of UV also generate late DSB after the conversion of base damage to SSB during the excision-resynthesis process and the conversion of SSB to DSB if base damage and/or SSB are too numerous [58]. Such succession from one DNA damage type to another requires some time and may explain the occurrence of late DSB. The presence of foci formed by the MRE11 nuclease that was found to be involved in base damage, SSB, and DSB repair supports such a hypothesis [59]. However, the most important clue of the concomitant presence of SSB and DSB is reflected by the “tiny” γH2AX foci that were systematically observed in RIBE cells. Some tiny γH2AX foci were also observed after UV exposure and treatments to H_2_O_2_ and disappeared after treatment with anti-oxidative drugs like NAC. Such observations strongly suggest that tiny γH2AX foci reflect the presence of some DSB (if not, no γH2AX signal would appear) and chromatin decondensation due to numerous SSB that disturb the γH2AX signals. Hence, tiny γH2AX foci in RIBE cells may reveal the existence of a strong oxidative stress, generating base damage, SSB, and DSB. A number of anti-oxidative drugs have been used to modulate RIBE [8,55]. This is notably the case for DMSO, catalase, and superoxide dismutase, that have been shown to inhibit RIBE, whether after micro-irradiation or medium transfer. In this study, data obtained with NAC suggest that the occurrence of RIBE is influenced by scavenging oxidative species and that RIBE DSB can be considered as “chemically” induced DSB [60].

### 4.3. Dependence of RIBE on Calcium and Phosphates

Differences in RIBE intensity observed between culture media as simple as PBS or physiological serum are impressive and fully eliminate any potential impact of bovine serum, antibiotics, or any specific complement present in the culture medium. Former observations have strongly suggested that extracellular flux of Ca^2+^ ions is a key event in RIBE [30]. Calcium plays a central role in cell viability and stress signaling notably with micromolar intracellular concentrations (a human fibroblast contains about 150 nM Ca^2+^) and millimolar concentrations in extracellular medium (DMEM medium contains 1.8 mM CaCl_2_) [61,62,63]. Hence, if all irradiated cells would release their intracellular free Ca^2+^ ions, significant RIBE would occur after irradiation of 10^4^ cells, at least, to produce more than 1.8 mM free Ca^2+^ ions. Although probably overestimated, this calculation is in agreement with the hypothesis that RIBE is strongly dependent on the number of irradiated cells [64]. Furthermore, the fact that a minimal radiation dose is sufficient to trigger extracellular Ca^2+^ ion release may explain why RIBE is not necessarily linearly dose-dependent [27,65]. Altogether, these conclusions invite us to carefully evaluate the irradiated tissue volumes and the corresponding number of cells irradiated for better quantifying RIBE.

Lastly, the calcium-dependence of RIBE has been generally supported through activation of signaling pathways leading to apoptosis [61]. To overcome cell cycle effects and genome instability, we used here confluent non-transformed human fibroblasts. Human fibroblasts are generally subjected to mitotic death and senescence rather than apoptosis [35]. Hence, with the cellular model chosen here, apoptosis does not appear as a predominant feature of RIBE [62,66,67].

### 4.4. RIBE and Its Contribution to the Radiation Dose: Differences between the Approaches Tested

Our findings show that the contribution of RIBE to the biological dose may depend on the irradiation conditions (Figure 10).

In the case of micro-irradiation, the cells that are directly targeted by irradiation received also the indirect calcium dependent RIBE. By contrast, cells at the edge of micro-tracks and further received only the radiation-induced Ca^2+^ ions release: RIBE likely obeys a Fick’s diffusion law [68], a function of the time and the distance from the impact. The RIBE is inasmuch intense as cells are closed to the micro-tracks (Figure 10). Under these conditions, the RIBE cells received a dose equivalent to the integral of the radiation induced Ca^2+^ ions release during the time of the experiment. The dispersion of γH2AX foci in cells far from the micro-tracks supports the idea that micro-irradiation may be considered as a series of successively infinitesimal medium transfers from the micro-tracks through medium (Figure 10). As shown in Figure 3 and Figure 4, the equivalent dose received by cells situated at 10 µm from the micro-tracks represents 5% of the initial dose (i.e., some fraction of Gy). Interestingly, this dose may belong to the range of doses generating the hypersensitivity to low dose phenomenon (HRS) [44,69,70]. At this stage, the RIBE induced by micro-irradiation may therefore produce deleterious effects in the surrounding tissues, notably if they are radiosensitive (HRS preferentially occurs in radiosensitive cells) [45] (Figure 10).

In the case of medium transfer, the receptor cells received only the radiation induced Ca^2+^ for a shorter period of emission (t1) than with micro-irradiation. As shown in Figure 1, the equivalent dose received by receptor cells represents less than 1% of the initial dose (i.e., in the mGy range if the initial dose is 2 Gy). Interestingly, this dose range belongs to the doses generating the hormesis phenomenon [45]. At this stage, the RIBE induced by medium transfer, may therefore produce a beneficial effect in the surrounding tissues, more particularly if the cells are radioresistant (hormesis preferentially occurs in radioresistant cells) [45] (Figure 10).

In the case of a standard irradiation (e.g., mimicking radiotherapy), the irradiated cells received both the direct ionizations and the radiation induced Ca^2+^ ion release in a homogenous manner. As shown in Figure 8, the calcium-dependent events may represent about 20% of the initial dose. Such contribution remains strongly dependent on the calcium content/number of the cells or tissues irradiated and on the homogeneity of the irradiation (particles vs. rays). The fact that the calcium-dependent contribution to the biological dose represents 20% required further investigations to better relate the assessment of the physical dose with a better evaluation of the biological one (Figure 10).

## 5. Conclusions

By applying both medium transfer and micro-irradiations approaches to radioresistant fibroblasts with 50 keV synchrotron X-rays, our findings suggest a temporal coincidence with the occurrence of RIBE DSB from 10 min to 4 h post-irradiation. By using different culture media, RIBE appears to be strongly calcium-dependent and the contribution of such specific events to the dose have been determined for medium transfer, micro-irradiations and irradiation mimicking a standard radiotherapy session. Our findings suggest that the RIBE contribution to the dose differs from the irradiation protocol and may result in beneficial or otherwise deleterious effects in surrounding tissues. The extent of the RIBE appears to be dependent on the number of cells irradiated. Further experiments are needed to ask whether/how the RIBE impacts the response of radiosensitive cells and how the calcium availability of cells/tissues interplays with this response.

## Figures and Tables

**Figure 1 biomolecules-13-00542-f001:**
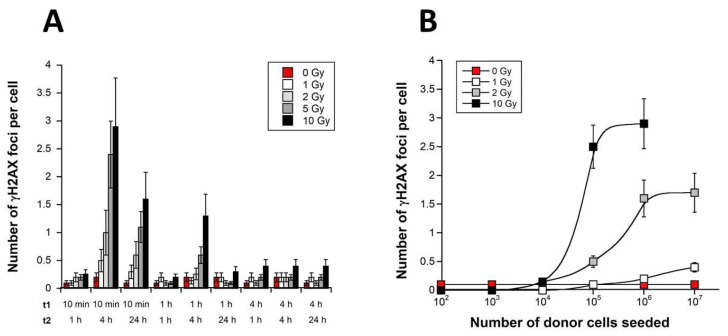
Characterization of the RIBE induced by a medium transfer approach with 50 keV synchrotron X-rays. (**A**) One million confluent 1BR3 donor fibroblasts were irradiated at the indicated doses with monochromatic 50 keV synchrotron X-rays in DMEM medium that was kept in the presence of donor cells for t1 post-irradiation (10 min–4 h), and then applied to unirradiated autologous receptor cells for t2 (1–24 h). The number of γH2AX foci per cell was plotted as a function of the t1 and t2 values. Each γH2AX data value represents the mean ± standard error of the mean (SEM) of three independent replicates. (**B**) By applying the same methodology, the number of γH2AX foci per cell was plotted against the number of donor cells irradiated at the indicated doses. Each γH2AX data value represents the mean ± SEM of three independent replicates. Data were fitted to a sigmoidal function.

**Figure 2 biomolecules-13-00542-f002:**
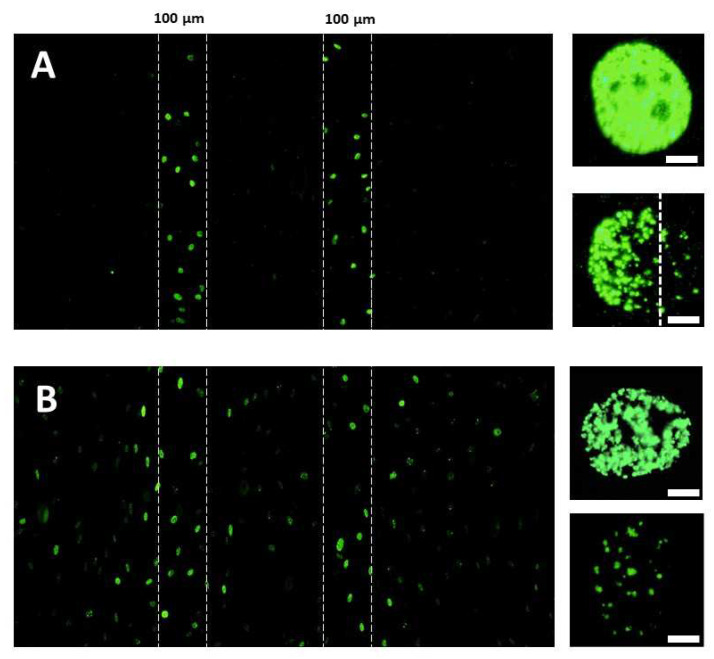
Characterization of the RIBE induced by a micro-irradiation approach with 50 keV synchrotron X-rays. Anti-*γH2AX* immunofluorescence was applied to the radioresistant 1BR3 fibroblasts irradiated at 10 Gy. Two 100-µm-wide parallel 50 keV synchrotron X-rays micro-tracks separated by 300 µm were generated (delimited by white dashed lines). Cells were fixed 10 min (**A**) or 4 h (**B**) post-irradiation and γH2AX immunofluorescence was applied (10× magnification). Inserts. Representative examples of γH2AX-positive nuclei situated either inside or at the edge of the micro-tracks (10 min post-irradiation) (**A**) or either inside or 200 µm far from the micro-tracks (**B**) (100× magnification). The white bars represent 5 µm.

**Figure 3 biomolecules-13-00542-f003:**
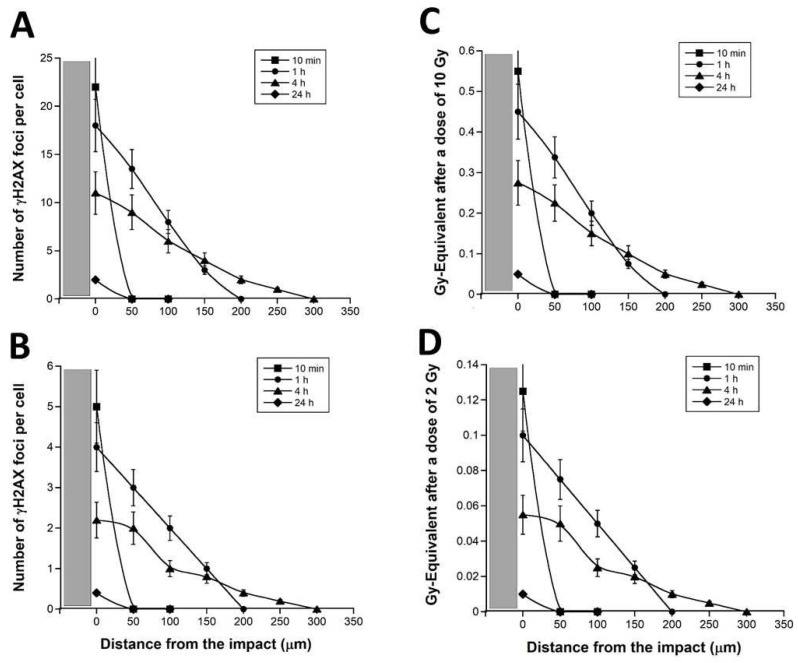
Spatial distribution of γH2AX foci in RIBE cells after a 50 keV synchrotron X-rays micro-irradiation. Anti-*γH2AX* immunofluorescence was applied to confluent 1BR3 cells subjected to X-rays micro-irradiation (10 Gy, **A**,**C** panels or 2 Gy, (**B**,**D**) panels) and cells were fixed at the indicated post-irradiation times. The number of γH2AX per RIBE cell was plotted against the distance from the micro-tracks. The grey zone represents a half micro-track. The number of γH2AX foci per cell shown in (**A**,**B**) panels was divided by 40 to express the corresponding data in Gy-equivalent ((**C**,**D**) panels). Each data plot represents the mean ± SEM of at least three replicates.

**Figure 4 biomolecules-13-00542-f004:**
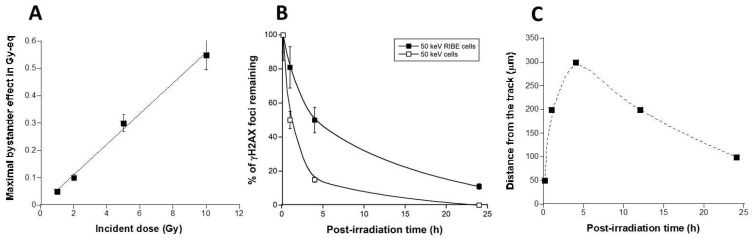
Specificities of the RIBE induced after a 50 keV synchrotron X-rays micro-irradiation in the 1BR3 fibroblasts. (**A**) Maximal RIBE expressed in Gy-equivalent (Gy-eq) as a function of dose (Gy) and deduced from the data shown in Figure 3C,D. These data were fitted to a linear formula (y = −0.00255 + 0.056 x; r = 0.997) (solid line). (**B**) RIBE DSB repair kinetics. The 10 µm γH2AX data shown in Figure 3 (2 Gy data) were plotted against the corresponding repair times. The resulting DSB repair kinetic of RIBE cells (closed squares) was compared with the DSB repair kinetic obtained with the same cell lines but from a standard (direct) 50 keV X-rays irradiation (open squares). These data were fitted to the Bodgi’s formula (solid lines) [47]. (**C**) From data shown in Figure 3, the distance from the track at which the number of γH2AX foci was found maximal was plotted against the corresponding post-irradiation time. Data were linked together with a smooth fit (dotted line).

**Figure 5 biomolecules-13-00542-f005:**
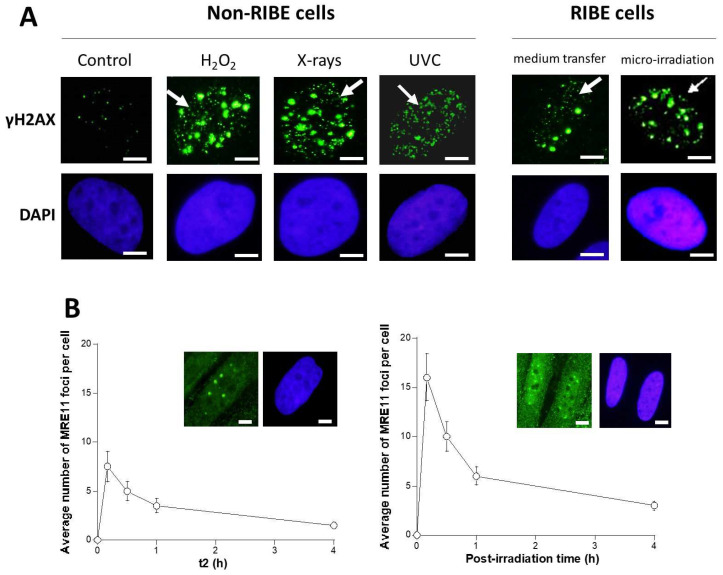
Tiny γH2AX foci as a specific feature of RIBE cells. (**A**) Different patterns of γH2AX foci observed in the radioresistant 1BR3 fibroblasts exposed to 10 mM H_2_O_2_ + 1 h, 1 Gy X-rays + 1 h, and 15 J.m^−2^ UVC + 1 h (left panel). Representative examples of RIBE nuclei after medium transfer (2 Gy; t1 = 10 min; t2 = 4 h) and micro-irradiation (10 Gy; 1 h) (right panel). White arrows indicate tiny foci. Nuclei were counterstained by DAPI. White bars represent 5 µm. (**B**) Average number of MRE11 foci per cell as a function of t2 (for the medium transfer approach after 2 Gy) or as a function of post-irradiation time (for the micro-irradiation approach after 2 Gy). Inserts are representative MRE11 immunofluorescence images assessed 1 h post-irradiation with DAPI counterstaining. White bars represent 5 µm.

**Figure 6 biomolecules-13-00542-f006:**
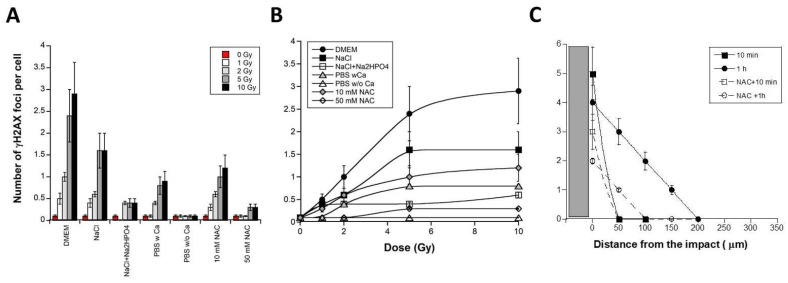
Effect of calcium on the RIBE. One million confluent 1BR3 fibroblasts were irradiated at the indicated doses with monochromatic 50 keV synchrotron X-rays by applying the medium transfer approach (t1 = 10 min and t2 = 4 h) with the indicated medium. Each data plot represents the mean ± SEM of at least three replicates. (**B**). The data shown in panel (**A**) were plotted as function of the initial dose. Each value represents the mean ± SEM of three independent replicates. (**C**). Spatial distribution of γH2AX foci in RIBE cells after monochromatic 50 keV synchrotron X-rays micro-irradiation. The data represented by closed symbols are those shown in the Figure 3B at 10 min and 1 h post-irradiation. Data represented by open symbols correspond to the *γH2AX* data in presence of 50 mM NAC. Each data plot represents the mean ± SEM of at least three replicates. In presence of PBSw/oCa, no γH2AX foci was observed (data not shown). The grey zone represents a half micro-track.

**Figure 7 biomolecules-13-00542-f007:**
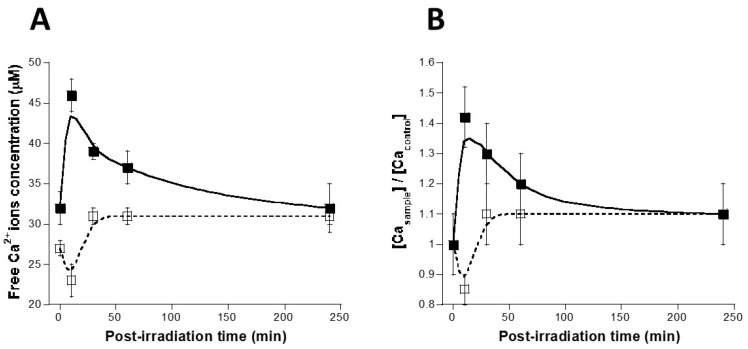
Calcium and RIBE. (**A**) Radiation-induced Ca^2+^ ions release in the radioresistant 1BR3 fibroblasts. One million 1BR3 fibroblasts were irradiated (10 Gy) in either PBSw/oCa (open squares; smooth fit with dotted line) or physiological serum (closed squares; smooth fit with closed line). Free Ca^2+^ ions concentration was determined in media 10, 30, 60, and 240 min post-irradiation. Each data plot represents the mean ± SEM of at least two replicates. (**B**) Data shown in panel (**A**) were normalised.

**Figure 8 biomolecules-13-00542-f008:**
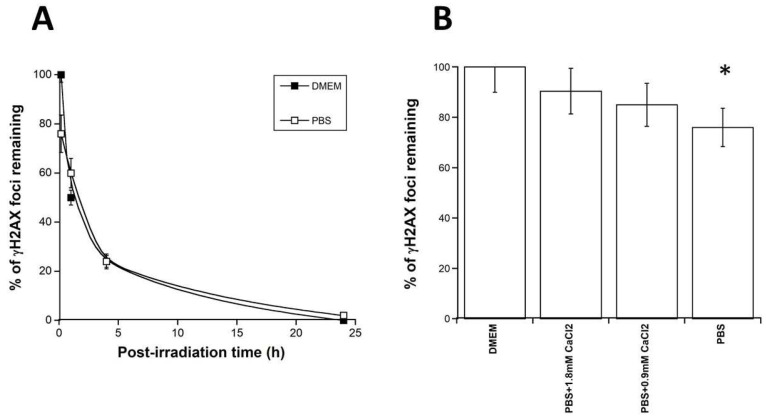
Influence of calcium on the DSB repair kinetics of the radioresistant 1BR3 fibroblasts. Cells were irradiated (2 Gy) with 6 MeV photons in presence of the indicated media. Each data plot represents the mean ± standard error of at least three replicates. (**A**) Kinetics of γH2AX foci. Data were fitted to the Bodgi’s formula [47]. (**B**) Histogram representing the γH2AX foci values (as percentage of foci remaining) assessed 10 min post-irradiation with the indicated media. Each data plot represents the mean ± SEM of at least three replicates. Asterisk represents *p* < 0.05.

**Figure 9 biomolecules-13-00542-f009:**
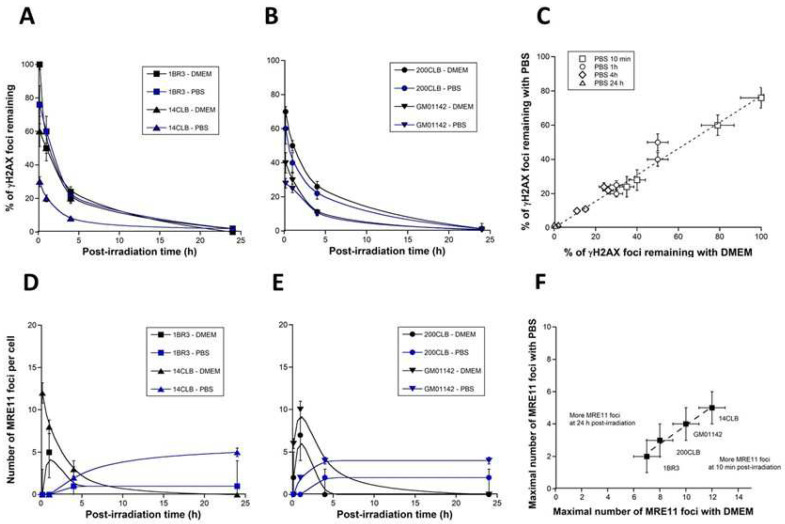
Calcium, RIBE and radiosensitivity. Th indicated four fibroblast cell lines with different radiosensitivity status were irradiated (2 Gy) with 6 MeV photons in presence of the indicated media (DMEM or PBS) and subjected to anti-*γH2AX* (**A**,**B**) or -*MRE11* (**D**,**E**) immunofluorescence at the indicated post-irradiation times. Each data plot represents the mean ± SEM of at least three replicates. Data were fitted to the Bodgi’s formula [47]. (**C**) The γH2AX data obtained with DMEM and PBS shown in panels (**A**,**B**) were plotted together with their corresponding error bars (SEM). The dotted line corresponds to the following formula: y = 0.75x (r = 0.99). (**F**) The MRE11 data obtained with DMEM and PBS shown in panels (**D**,**E**) from the indicated four fibroblast cell lines were plotted together with their corresponding error bars (SEM). The dotted line corresponds to the following formula: y = −1.83 + 0.576 × (r = 0.989).

**Figure 10 biomolecules-13-00542-f010:**
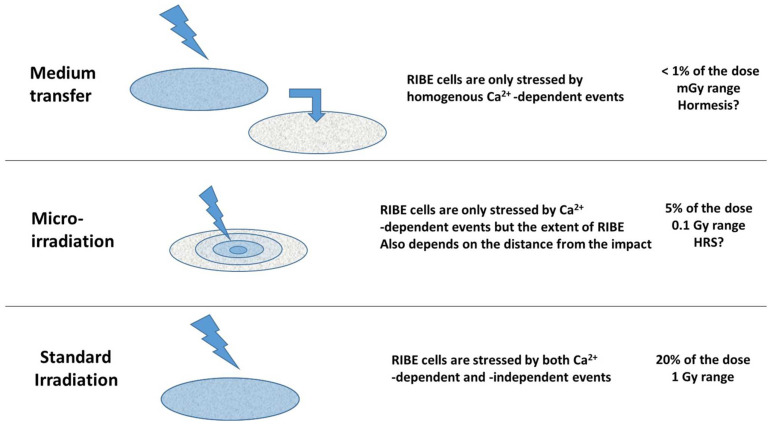
Schematic summary of the RIBE according to the irradiation protocol. For medium transfer, micro-irradiation and “standard” irradiation, all or a subset of cells are concerned by calcium-dependent events leading to RIBE DSB. For each protocol, the calcium-dependent events contribute differentially to the total dose and some specific radiobiological phenomena may occur (hormesis or HRS).

**Table 1 biomolecules-13-00542-t001:** Concentration (mM) of phosphates and calcium in the culture media used here.

	NaCl (0.9%)	DMEM	PBS and PBSw/oCa
NaCl	153	110.34	137.93
KCl	0	5.33	2.67
Na_2_HPO_4_	0	0.916	8.06
KH_2_HPO_4_	0	0	1.47
Phosphates	0	~1	~10
Calcium (CaCl_2_)	0	1.8	0.9 (for PBS)
RIBE-induced DSBs	+	+	-

## Data Availability

All the data can be provided on reasonable request.

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
