# Peer review of "X-rays-Induced Bystander Effect Consists in the Formation of DNA Breaks in a Calcium-Dependent Manner: Influence of the Experimental Procedure and the Individual Factor"

_biomolecules, 2023, doi:10.3390/biom13030542_

Round 1
Reviewer 1 Report
This is a beautifully designed and executed study which draws together a large number of disparate facts about bystander effects (RIBE) and unifies them into a very neat overall theory which suggest a key role for calcium ions and DSBs in the overall mechanism irrespective of how RIBE is induced.
Author Response
Reply to the reviewer 1
This is a beautifully designed and executed study which draws together a large number of disparate facts about bystander effects (RIBE) and unifies them into a very neat overall theory which suggest a key role for calcium ions and DSBs in the overall mechanism irrespective of how RIBE is induced.
We thank this reviewer for his/her comments
We must apologize because it appears that some paragraphs of the first version were not edited equally. See modified text in all the manuscript
Reviewer 2 Report
The manuscript entitled “X-rays-induced bystander effect consists in the formation of DNA breaks in a calcium-dependent manner: influence of the experimental procedure and the individual factor” compared responses and possible mechanism(s) underlying the medium transfer and microirradiation approaches using the same kind of irradiation source-the synchrotron X-rays. Further, the signal of MRE11 foci, which recognizes SSBs and DSBs that are not managed by NHEJ was evaluated in addition to the commonly utilized γH2AX foci to dissect the RIBE phenomenon more comprehensively. The study provided interesting information concerning RIBE that originated from different experimental procedures with the same kind of irradiation source. However, there are quite a few critical issues that still need to be addressed:
1) As indicated in the title, different experimental procedures (influence of the experimental procedure and the individual factor) seem to affect RIBE. Such kind of comparisons need to be more fully elaborated in the manuscript.
2) It seems to be a problem to this reviewer that the authors used the percentage of cells showing more than 2 γH2AX foci as a molecular endpoint for RIBE. The phenomenon that more cells show γH2AX foci means that DNA damage occurs in more receptor cells, which also suggests the occurrence of RIBE. However, the foci number in each of the cells may not necessarily be higher than those occur in spontaneous events.
3) Does the statement “lower than 2 γH2AX foci” means two γH2AX foci per cell? As “the percentage of cells showing more than 2 γH2AX foci was taken as a molecular endpoint in the medium transfer experiments”, why a different biological endpoint in that the number of γH2AX foci per cell was utilized for the microirradiation study? It seems more logical and scientifically sound to use the same endpoint for the comparison of these two kinds of treatments.
4) In the case of microirradiation, the number of cells seeded should be provided. Moreover, a bright field of the microscope observation showing cell monolayer should be provided to clarify how close the bystander cells are related to the irradiated cells. As it is well known that two general mechanisms are underlying RIBE- one mediated by gap junction and one mediated by soluble factors in the conditioned medium. The confluence of cells, therefore, is critical in data interpretation.
5) On page 8, lines 280-281, it was said that “Such statement remains true for 1,2, 5 and 10 Gy (data not shown)”, then what is the dose shown in Figure 4B&C?
6) Calcium is more generally considered as a secondary messenger and hence the change of intracellular concentration of calcium is more important for signal transduction and should be measured instead. The detection of up to 40 uM change of free calcium in the medium (Figure 7A) seems to be overwhelming. Whether the cells were grown in a calcium-free (and phosphate-rich) medium for a prolonged period of time before irradiation should be indicated. As the presence or absence of an intracellular calcium pool in the irradiated and/or bystander cells is important for a proper interpretation of the results obtained.
7) In Figure 7C, “the calcium data shown in panel A were plotted against the corresponding number of MRE11 foci shown in fig. 5 for the same post-irradiation time”. However, figure 7A showed the results from irradiation with 10 Gy, while in figure 5B, the dose used for obtaining the MRE11 foci data was indicated as 2 Gy. How data obtained from different doses can be plotted in the same figure needs to be explained. Also in this figure, how data from only two replicates can result in a standard error is unclear.
8) It is not clear how RIBE was observed in Figures 8&9. Were these experiments performed with the medium transfer or microirradiation method? Or the γH2AX foci were measured in the directly irradiated cells? In addition, plotting all the cell lines in one figure (such as that in Figures 9A&C) made it very hard to evaluate the quality of the data. The data presented in Figure 9B and D are quite confusing. What kind of cell lines was presented in Figure 9B and what each of the symbols stood for should be explained more clearly. Similar problems were found in Figure 9D.
9) English needs to be improved. Many sentences are hard to understand. For example, on page 6, lines 204-206, “In our conditions, a maximal RIBE was observed at doses higher than 2 Gy and with more than 106 cells seeded (Fig. 1B)”. The maximal RIBE should be corresponding to a single condition, while “doses higher than 2 Gy and cell numbers higher than 106” stand for a range of conditions. Therefore, the meaning of the sentence is ambiguous. Many typos are found in the manuscript as well. For example, in the title for Figure 4, microirradiationo should be microirradiation; page 11, line 374, “data nt shown” should be data not shown; page 18, line 583, over-restimated should be over-estimate, etc.
Author Response
Reply to the reviewer 2
We thank the reviewer for his/her comments.
We must apologize because it appears that some paragraphs of the first version were not edited equally. See modified text in all the manuscript
The manuscript entitled “X-rays-induced bystander effect consists in the formation of DNA breaks in a calcium-dependent manner: influence of the experimental procedure and the individual factor” compared responses and possible mechanism(s) underlying the medium transfer and microirradiation approaches using the same kind of irradiation source-the synchrotron X-rays. Further, the signal of MRE11 foci, which recognizes SSBs and DSBs that are not managed by NHEJ was evaluated in addition to the commonly utilized γH2AX foci to dissect the RIBE phenomenon more comprehensively. The study provided interesting information concerning RIBE that originated from different experimental procedures with the same kind of irradiation source. However, there are quite a few critical issues that still need to be addressed:
- As indicated in the title, different experimental procedures (influence of the experimental procedure and the individual factor) seem to affect RIBE. Such kind of comparisons need to be more fully elaborated in the manuscript.
OK see abstract and introduction that have been deeply modified
- It seems to be a problem to this reviewer that the authors used the percentage of cells showing more than 2 γH2AX foci as a molecular endpoint for RIBE. The phenomenon that more cells show γH2AX foci means that DNA damage occurs in more receptor cells, which also suggests the occurrence of RIBE. However, the foci number in each of the cells may not necessarily be higher than those occur in spontaneous events.
We have modified the figures 1 and 6 that now show the number of γH2AX foci per cell
Does the statement “lower than 2 γH2AX foci” means two γH2AX foci per cell? As “the percentage of cells showing more than 2 γH2AX foci was taken as a molecular endpoint in the medium transfer experiments”, why a different biological endpoint in that the number of γH2AX foci per cell was utilized for the microirradiation study? It seems more logical and scientifically sound to use the same endpoint for the comparison of these two kinds of treatments.
OK You are perfectly right. We have modified the figures 1 and 6 that now show the number of γH2AX foci per cell
In the case of microirradiation, the number of cells seeded should be provided. Moreover, a bright field of the microscope observation showing cell monolayer should be provided to clarify how close the bystander cells are related to the irradiated cells. As it is well known that two general mechanisms are underlying RIBE- one mediated by gap junction and one mediated by soluble factors in the conditioned medium. The confluence of cells, therefore, is critical in data interpretation.
OK see new figure in supplementary data with a representative image (bright field) of fibroblasts monolayers
On page 8, lines 280-281, it was said that “Such statement remains true for 1,2, 5 and 10 Gy (data not shown)”, then what is the dose shown in Figure 4B&C?
You are right. Data concerned were those obtained at 2 Gy. See modified text and legend.
Calcium is more generally considered as a secondary messenger and hence the change of intracellular concentration of calcium is more important for signal transduction and should be measured instead. The detection of up to 40 uM change of free calcium in the medium (Figure 7A) seems to be overwhelming. Whether the cells were grown in a calcium-free (and phosphate-rich) medium for a prolonged period of time before irradiation should be indicated. As the presence or absence of an intracellular calcium pool in the irradiated and/or bystander cells is important for a proper interpretation of the results obtained.
See modified text materials and methods
In Figure 7C, “the calcium data shown in panel A were plotted against the corresponding number of MRE11 foci shown in fig. 5 for the same post-irradiation time”. However, figure 7A showed the results from irradiation with 10 Gy, while in figure 5B, the dose used for obtaining the MRE11 foci data was indicated as 2 Gy. How data obtained from different doses can be plotted in the same figure needs to be explained. Also in this figure, how data from only two replicates can result in a standard error is unclear.
You are right. We preferred to delete the Fig. 7C
It is not clear how RIBE was observed in Figures 8&9. Were these experiments performed with the medium transfer or microirradiation method? Or the γH2AX foci were measured in the directly irradiated cells?
Yes in directly irradiated cells as the modified text described it.
In addition, plotting all the cell lines in one figure (such as that in Figures 9A&C) made it very hard to evaluate the quality of the data. The data presented in Figure 9B and D are quite confusing. What kind of cell lines was presented in Figure 9B and what each of the symbols stood for should be explained more clearly. Similar problems were found in Figure 9D.
OK See modified and new figures
English needs to be improved. Many sentences are hard to understand. For example, on page 6, lines 204-206, “In our conditions, a maximal RIBE was observed at doses higher than 2 Gy and with more than 106cells seeded (Fig. 1B)”. The maximal RIBE should be corresponding to a single condition, while “doses higher than 2 Gy and cell numbers higher than 106” stand for a range of conditions. Therefore, the meaning of the sentence is ambiguous. Many typos are found in the manuscript as well. For example, in the title for Figure 4, microirradiationo should be microirradiation; page 11, line 374, “data nt shown” should be data not shown; page 18, line 583, over-restimated should be over-estimate, etc.
We must apologize because it appears that some paragraphs of the first version were not edited equally. See modified text in all the manuscript
Reviewer 3 Report
Summary and revisions:
My impression of this manuscript is that it is an important contribution to the field of radiation bystander biology, with limited importance and application. I would like for the authors to consider re-organizing the sections for better flow of information. The manuscript is very dense with information, however, the good stuff is lost within the conversational style of writing. There are lot of inferences made based on numbers with little statistics shown/mentioned. The readers need to know that the data are being presented are statistically significant and this does not come across here. Additionally, the manuscript needs a major revision for correct English and grammar. Many words are misplaced or misused, there are too many to list here. I list a couple of minor points below, but after continuing to read the manuscript, there were too many typos and confusing points to keep listing. I strongly suggest that the authors rework the paper because the science is strong, just the presentation needs to be majorly improved.
Minor revisions:
1. Neither ref 7 or 8 have gene expression results, there are many other papers that can be referenced here.
2. The authors mention that the use of synchrotron xrays is advantageous but not in what context. This is an important concept that needs to be further elucidated.
3. For all figures, the text is too small even in the legends, there is too much white space and the lines and points are not different enough to be clearly seen.
Author Response
We thank this reviewer for his/her comments
We must apologize because it appears that some paragraphs of the first version were not edited equally. See modified text in all the manuscript
Summary and revisions:
My impression of this manuscript is that it is an important contribution to the field of radiation bystander biology, with limited importance and application. I would like for the authors to consider re-organizing the sections for better flow of information. The manuscript is very dense with information, however, the good stuff is lost within the conversational style of writing. There are lot of inferences made based on numbers with little statistics shown/mentioned. The readers need to know that the data are being presented are statistically significant and this does not come across here.
The text has been deeply modified.
Additionally,the manuscript needs a major revision for correct English and grammar. Many words are misplaced or misused, there are too many to list here. I list a couple of minor points below, but after continuing to read the manuscript, there were too many typos and confusing points to keep listing. I strongly suggest that the authors rework the paper because the science is strong, just the presentation needs to be majorly improved.
The text has been deeply modified and the English has been reviewed. We apologize since it appears that some parts of the text were not taken from the very last version of our manuscript.
Minor revisions:
- Neither ref 7 or 8 have gene expression results, there are many other papers that can be referenced here.
OK. see modified references and text.
- The authors mention that the use of synchrotron xrays is advantageous but not in what context. This is an important concept that needs to be further elucidated.
OK. See modified text in Introduction
- For all figures, the text is too small even in the legends, there is too much white space and the lines and points are not different enough to be clearly seen.
OK. See modified figures
Reviewer 4 Report
The authors investigated the radiation-induced bystander effects (RIBE) and various experimental procedures have been used. The study is interesting and well written including the description of the methods employed. The discussion is very clear and the implications for the observed phenomena have been adequately considered.The results are quite important in the fields of radiobiologists for understanding the molecular basis involved in RIBE. Some specific and minor suggestions are reported below:
The Introduction section gives an idea of what hypotheses the authors want to address.
Materials and methods:
Please, insert all reagents with the same manufacturer’s once (eg. reagents from Sigma-Aldrich).
How many cells have been observed for foci scoring?
In figure 1 the authors should be indicated the statistic significant effects observed and the same for other figures when needed.
Are the experiments always three independent for each experimental run?
The biological part was adequately presented and supported the conclusions.
The work is adequately presented in the context of the available literature.
Author Response
Reply to the reviewer 4
We thank this reviewer for his/her comments
We must apologize because it appears that some paragraphs of the first version were not edited equally. See modified text in all the manuscript
The authors investigated the radiation-induced bystander effects (RIBE) and various experimental procedures have been used. The study is interesting and well written including the description of the methods employed. The discussion is very clear and the implications for the observed phenomena have been adequately considered. The results are quite important in the fields of radiobiologists for understanding the molecular basis involved in RIBE. Some specific and minor suggestions are reported below:
The Introduction section gives an idea of what hypotheses the authors want to address.
The Introduction has been deeply modified. See modified text.
Materials and methods:
Please, insert all reagents with the same manufacturer’s once (eg. reagents from Sigma-Aldrich).
See modified text
How many cells have been observed for foci scoring?
See modified in Materials and Methods (Immunofluorescence section).
In figure 1 the authors should be indicated the statistic significant effects observed and the same for other figures when needed.
See modified figures
Are the experiments always three independent for each experimental run?
No. See the captions of each figure.
The biological part was adequately presented and supported the conclusions.
The work is adequately presented in the context of the available literature.
Round 2
Reviewer 2 Report
The quality of the revised version of the manuscript “X-rays-induced bystander effect consists in the formation of DNA breaks in a calcium-dependent manner: influence of the experimental procedure and the individual factor” is greatly improved. The authors have addressed most of the questions raised by this reviewer. However, What kind of cell lines was presented in Figure 9B (now Figure 9C) and what each of the symbols stood for are still not clear.
Responses to the reviewer
Thank you very much for your comments. We have defined the cell line and symbols, please see the modified manuscript.